# Addressing missing data in randomized clinical trials: A causal inference perspective

Ilja Cornelisz[1], Pim Cuijpers[2,3], Tara Donker[2,3], Chris van Klaveren[1] *

**1** Amsterdam Center for Learning Analytics, Vrije Universiteit Amsterdam, Amsterdam, The Netherlands, **2** Department of Clinical, Neuro- and Developmental Psychology, Section Clinical Psychology, Vrije Universiteit Amsterdam, Amsterdam, The Netherlands, **3** Amsterdam Public Health Research Institute, Amsterdam, The Netherlands

* c.p.b.j.van.klaveren@vu.nl

**Data Availability Statement:** The data used for the paper is uploaded as a STATA file, as well as the STATA Code that generates all the results. We will

## Abstract

### Background

The importance of randomization in clinical trials has long been acknowledged for avoiding selection bias. Yet, bias concerns re-emerge with selective attrition. This study takes a causal inference perspective in addressing distinct scenarios of missing outcome data (MCAR, MAR and MNAR).

### Methods

This study adopts a causal inference perspective in providing an overview of empirical strategies to estimate the average treatment effect, improve precision of the estimator, and to test whether the underlying identifying assumptions hold. We propose to use Random Forest Lee Bounds (RFLB) to address selective attrition and to obtain more precise average treatment effect intervals.

### Results

When assuming MCAR or MAR, the often untenable identifying assumptions with respect to causal inference can hardly be verified empirically. Instead, missing outcome data in clinical trials should be considered as potentially non-random unobserved events (i.e. MNAR). Using simulated attrition data, we show how average treatment effect intervals can be tightened considerably using RFLB, by exploiting both continuous and discrete attrition predictor variables.

### Conclusions

Bounding approaches should be used to acknowledge selective attrition in randomized clinical trials in acknowledging the resulting uncertainty with respect to causal inference. As such, Random Forest Lee Bounds estimates are more informative than point estimates obtained assuming MCAR or MAR.

also make the data publicly available at acla. amsterdam.

**Funding:** The author(s) received no specific funding for this work

**Competing interests:** The authors have declared that no competing interests exist.

## Background

Empirical research is considered most valuable when it aims to answer a specific causal question [1], as is the case in a randomized clinical trial. Randomization has the objective to avoid potential bias in the estimation of causal treatment effects by ensuring that a participant's treatment status is independent from both observed and unobserved characteristics. If successful, randomization enables internal validity [2], such that differences between participants in a clinical trial assigned to different treatment arms can, apart from random error, be attributed to the treatment under investigation [3].

However, randomized clinical trials often deploy some form of restriction when assigning participants to treatment arms, for example blocking, stratification or minimization, as to improve the balance obtained on known confounders [4]. Such restrictions increase the risk of subversion whenever random allocation sequences are observed or can be–partially- predicted; introducing pathways for selection bias in a randomized trial [5]. From a causal inference perspective, simple randomization is then to be preferred to avoid introducing such bias; thereby sacrificing some level of precision (i.e. resulting from covariate imbalances).

Yet, even if initial randomization is successful, additional concerns regarding causal inference arise in the presence of missing data. Missing data problems can introduce subsequent– and severe- bias when analyzing data from randomized clinical trials and should thus be acknowledged empirically. Several dimensions of missing data exist, which differ considerably in terms of their potential for raising causal inference concerns. In this paper, we focus on these phenomena and consider missing data on either background characteristics, treatment status or outcomes.

For causal inference, it is important to empirically acknowledge the potential importance of missing data problems. These issues can introduce bias in subsequent data analyses, thereby undermining initial randomization. Several dimensions of missing data exist, which differ considerably in terms of potential causal inference concerns. We consider missing data on background characteristics, treatment status and outcomes.

Missing data on background characteristics can be addressed relatively easy, either by means of a missing indicator method or imputation, thereby rendering the sample of randomized participants complete. Given that background characteristics are independent from treatment status by virtue of randomization, the resulting causal inferences with respect to the average treatment effect are internally valid.

Missing data on treatment status occurs when participants discontinue, for whatever reason, the assigned treatment. When gathering data on outcomes for these participants is successful, the possibility to estimate an internally valid intent-to-treat effect is retained. This measure–in itself- is informative regarding treatment policy effects and provides a lower-bound estimate of the average treatment effect.

Missing data on outcomes is widespread in clinical trials [6]. If such attrition co-varies with (un)observed characteristics and treatment assignment, this compromise initial randomization and causal inferences drawn from clinical trials [7]. While attrition bias may thus render the results from a study non-informative, the problem of missing outcome data has received relatively little attention in the clinical trial literature [8]. Missing data is often inadequately addressed by means of simple fixes [9], but also more advanced methods such as multiple imputation (MI) can yield biases as big as, or even bigger than, the bias in simple complete cases analysis results [8].

Based on a widely established categorization of missing outcome data scenarios [10], this study proposes empirical strategies for each scenario in the context of a randomized clinical trial and by adopting a causal inference perspective. An empirical solution for obtaining an

internally valid treatment effect estimate in the presence of missing data is provided for each scenario. In addition, the underlying identifying assumptions that apply are made explicit, together with the extent to which–and how- they can be verified empirically.

Three missing data scenarios are distinguished, and each imposes different assumptions with respect to characterizing the missing data generation process:

1. Missing Completely At Random (MCAR)

2. Missing At Random (MAR)

3. Missing Not At Random (MNAR)

MCAR assumes that the occurrence of a missing outcome is a random event, unrelated to treatment assignment and–observed and unobserved- background characteristics. MAR assumes that observed background characteristics can account for the non-random nature of missing data. Finally, MNAR assumes that the occurrence of a missing outcome is a non-random event that can be structurally related to both observed and–importantly- unobserved background characteristics. From a causal inference perspective, each scenario entails different empirical strategies to test its underlying identifying assumptions and also to address missing data concerns.

## Addressing selective attrition

This study argues that -from a causal inference perspective- it is most appropriate to assume that the missing outcome data generating process in a randomized clinical trial is MNAR. Both MCAR and MAR impose overly restrictive identifying assumptions that are crucial for claiming that the estimated effect is internally valid, yet often unrealistic and empirically unverifiable. In the case of MNAR, common empirical bounding approaches rely on exploring worst-case scenario bounds, often resulting in very wide, uninformative, treatment effect estimate intervals.

Therefore, this study proposes to use a new Random Forest Lee Bounds (RFLB) approach that can be used to estimate relatively tight average treatment effect intervals, when assuming MNAR. The potential of exploiting RFLB in the presence of selective attrition is demonstrated using data from a recent randomized clinical trial which established the treatment effect of virtual reality cognitive behavior therapy (VR CBT) in addressing acrophobia (Donker et al., 2019). We show how RFLB can considerably increase the statistical power of bias-corrected treatment effect estimates for randomized clinical trials facing non-random missing outcome data, by exploiting both continuous and discrete predictor variables for attrition.

## Baseline model

Assume a randomized clinical trial conducted with the objective to estimate the effect of treatment $T$ on outcome $y_i$ for individual $i$ (with $i = 1,. . .,N$). The two potential outcomes for individual $i$ can then be represented by:

$$y_i = \begin{cases} y_{1i} & \text{if } T_i = 1 \\ y_{0i} & \text{if } T_i = 0 \end{cases},$$

and the individual treatment effect for individual $i$ as:

$$y_{0i} + (y_{1i} - y_{0i})T_i.$$

Individuals cannot be randomly assigned to the treatment and control group simultaneously, such that only one of both potential outcomes $y_{0i}$ and $y_{1i}$ is observed for each individual which prevents the estimation of the individual treatment effects [2]. This observation is referred to as the fundamental problem of causal inference. Yet, the average treatment effect for these individuals can be estimated empirically using ordinary least squares regression:

$$y_i = \alpha + \beta \cdot T_i + \varepsilon_i, \tag{1}$$

where $\alpha$ and $\beta$ are the parameters to be estimated and where $\varepsilon$ represents the classical error term, assumed to be identically and independently distributed (i.i.d) with mean zero ($E(\varepsilon_i) = 0$). The estimated expected outcome for treated and non-treated individuals is $E(y_i|T = 1) = \alpha + \beta$ and $E(y_i|T = 0) = \alpha$, respectively, yielding the difference between these expected outcomes, $\beta$, to represent an unbiased estimated average treatment effect. Performing an independent sample t-test would yield the same average treatment effect.

A more precise estimate of this treatment effect can be obtained empirically by including observed background characteristics that are related to $y_i$ in Eq 1:

$$y_i = \alpha + \beta \cdot T_i + \boldsymbol{X_i'\delta} + \varepsilon_i. \tag{2}$$

Vector $\mathbf{X}$ represents the vector of observed background characteristics and parameter vector $\boldsymbol{\delta}$ are the associated model parameters to be estimated. The estimated treatment effect is unbiased in both Eqs 1 and 2, but the inclusion of background characteristics that are significantly related with the outcome variable reduces the standard error of $\beta$ and thus increases the statistical power of empirically detecting this treatment effect.

## Results

### Missing completely at random (MCAR)

MCAR assumes that missing outcomes are unrelated with treatment status and *all* background characteristics, whether observed or unobserved. Let indicator variable $M$(issing) denote 1 when outcome $y_i$ is missing, and 0 otherwise. This *conditional mean zero* (CMZ) assumption of MCAR then implies $E(\varepsilon_i|T_i,\boldsymbol{X_i}) = 0$, meaning that both observed treatment $T_i$ and background characteristics $\mathbf{X}$ are uncorrelated with the error term $\varepsilon_i$.

Let $N_{fs}$ and $N_{os}$ be the number of observations for, respectively, the full sample and the sample of individuals for which an outcome is observed and let $N_{os} \leq N_{fs}$. If the occurrence of missing outcomes is truly random, then $P(T|M) = P(T)$ and an unbiased estimator of the treatment effect can be obtained empirically by estimating either Eq 1 or–for precision reasons- Eq 2.

The analysis can then be performed on the 'complete case' sample, also referred to as listwise deletion. A drawback of this procedure is that the use of the smaller sample $N_{os}$ can severely reduce statistical power. More precisely, if the goal was to have 80% statistical power to empirically detect a treatment effect, this power reduction by using the smaller sample $N_{os}$ is reflected by $\theta$, for which holds (see S1 Appendix for a full derivation of $\theta$):

$$\theta = 2.8 \cdot \sqrt{\frac{N_{os}}{N_{fs}}} - 1.96. \tag{3}$$

A power of 80% is associated with a $\theta$-value of 0.84 (i.e. $N_{os} = N_{fs}$). Eq 3 illustrates that the statistical power is reduced by attrition as $N_{os} < N_{fs}$. For example, when $\sqrt{\frac{N_{os}}{N_{fs}}} = 0.75$, this implies a value for $\theta$ of 0.14, yielding a power of 56% and–thus- a power reduction of 24 percentage points (see S1 Appendix).

Therefore, a-priori knowledge about the extent to which missing outcomes will occur should be used to make an educated guess on the full-sample size needed ($N_{fs}$), as to retain the ability to perform a complete case analysis on the observed sample ($N_{os}$) with 80% statistical power.

A more fundamental problem with *MCAR* is that the randomness assumption of missing outcomes (CMZ) cannot be demonstrated, but merely be assumed. Empirically, this assumption can only be partially validated by two different estimation strategies. The first strategy is to estimate a (logistic/probalistic) regression model and show that treatment status $T_i$ cannot be predicted empirically by the set of observed background characteristics **X**:

$$T_i = \alpha + X_i' \delta + \varepsilon_i. \tag{4}$$

Generally a logistic or probability model is estimated, given the dichotomous nature of the dependent variable. Yet, for the validation purpose of causal inference considered here a simple linear regression model is sufficient to estimate, as we are not interested in the interpretation of the estimation parameters. The observed background characteristics should not explain treatment status if the identifying assumption of *MCAR* holds and the estimated coefficients of vector $\delta$ should all be statistically insignificant and close to zero. A second estimation strategy is to estimate–and compare- Eqs 5 and 6:

$$y_i = \alpha + \beta \cdot T_i + \varepsilon_i. \tag{5}$$

$$y_i = \alpha + \beta \cdot T_i + X_i' \delta + \varepsilon_i. \tag{6}$$

The estimated parameter $\beta$ in the Eq 5 provides the t-test or Wald-test estimator of the treatment effect. The inclusion of observed background characteristics in Eq 6 should not statistically significantly change the estimated treatment effect $\beta$ if the MCAR assumption of CMZ holds. The logic is similar to that of the first estimation strategy: if it is assumed that *P(T|M) = P(T)*, then *P(T|M, X) = P(T)* must also hold. However, if missing outcomes are structurally related to relevant background characteristics, then the estimated treatment effect -or treatment assignment- will depend on whether these background characteristics are included or not.

A key insight from both strategies is that showing that $T_i$ is independent from observable characteristics is sufficient to argue this will also hold for all unobserved characteristics. The estimated treatment effect $\beta$ can properly be inferred causally if this identifying assumption holds, but this can thus merely be assumed and not demonstrated unequivocally.

Based on the above, it could be argued that the term missing completely at random is somewhat confusing. It suggests that missing outcomes are not problematic due to its random nature, yet it is precisely this random nature that cannot be proven. At best, one can show empirically that the estimated treatment effect is independent from observed characteristics and–then- to assume that this will also be the case for all unobserved characteristics.

## Missing At Random (MAR)

Different from MCAR, MAR assumes that missing outcomes can actually be non-random in nature, such that the CMZ not hold and Eq 5 will yield a biased treatment effect estimate. Yet, with MAR it is assumed that the selective nature of missing outcomes can fully be accounted for by a set of observed background characteristics **X**. This implies that P*(T|M, X) = P(T|X)*, such that treatment status $T_i$-once controlled for observed background characteristics- is unrelated to missing observation status. This identifying assumption is referred to as *conditional mean independence* (CMI) and implies $E(\varepsilon_i|T_i, X) = E(\varepsilon_i|X) \neq 0$. This is unproblematic as long as

we are only interested in inferring the causal effect of treatment $T_i$ and not in the causal effect of background characteristics $\mathbf{X}$. Conditional on X, $T_i$ is as if randomly assigned, such that treatment $T_i$ is uncorrelated with $\varepsilon_i$, but $\mathbf{X}$ can be correlated with $\varepsilon_i$. The CMI assumption then implies that $E(\varepsilon_i|T_i = 1,\mathbf{X}) = E(\varepsilon_i|T_i = 0,\mathbf{X})$, yielding $E(y_i|T = 1,\mathbf{X})-E(y_i|T = 0, \mathbf{X}) = \beta$ and thus that the treatment effect estimate from Eq 6 is unbiased.

Whereas *MAR* is different from *MCAR* in assuming *conditional* randomness with respect to the missing outcomes, the underlying identifying assumption necessary to argue that the estimated treatment effect is causal is similar for both scenarios. As with *MCAR*, the validity of the identifying assumption for MAR–conditional mean independence- cannot be demonstrated. The two estimation strategies outlined before with *MCAR* can again be pursued, but are this time performed to show (1) how the probability of The pivotal assumption underlying *MAR* is that the inclusion of the set of observed background characteristics $\mathbf{X}$ in Eq 6 is sufficient to also account for the potential effects of non-observed background characteristics on observed treatment status. This assumption is similar to situation described in *MCAR*, and–again- it is not possible to empirically validate this conditional mean independence assumption on non-observed background characteristics. In practical terms, this implies that *MCAR* and *MAR* are similar in the identifying assumptions made, with the notable exception that the inclusion of observed background characteristics in Eq 5 can alter the estimated treatment effect for *MAR*, while this is not allowed to be possible with *MCAR* (and are only included for precision).

The aforementioned observation that the estimated treatment effect is allowed to change with *MAR* due to the inclusion of $\boldsymbol{X}$ is what makes it distinctively different from *MCAR*. Whereas the estimation strategy in Eq 5 with *MCAR* serves to falsify its identifying assumption of missing outcomes as purely random events, it is used with *MAR* to empirically control for the non-random nature of missing outcomes. The identifying assumption for *MAR* that treatment status is then CMI on unobservables -after having included the observed characteristics $\mathbf{X}$- is a strong assumption to make and empirically untractable.

## Missing Not At Random (MNAR)

Given that randomness of missing outcomes cannot be demonstrated, a more appropriate point of departure is MNAR and to assume that missing outcomes *can* result from non-random events (Graham, 2009), such that the probability of observing a missing outcome can be structurally related to both observed and–importantly- unobserved characteristics. In contrast to MAR, MNAR thus implies that the inclusion of observed background characteristics $\mathbf{X}$ cannot control for the selective nature of missing values, rendering results thus obtained to be biased.

MNAR thus not assume CMZ or CMI, such that no unbiased point estimate for $\beta$ can be generated using Eq 5 or 6. As such, additional assumptions regarding the potential selective nature of missing outcomes are required to estimate $\beta$. Empirically, this often implies performing a bounding procedure in which two contrasting scenarios regarding attrition (e.g. "worst"- and "best"-case) are considered; thereby yielding an interval estimate for $\beta$ instead.

**Bounding procedures.**  Rather than correcting the point estimate of treatment effect $\beta$ for potential bias, bounding procedures yield treatment effect interval estimates instead. In their seminal work on dealing with missing data in randomized experiments, Horowitz and Manski [11] first provide a general assumption-free framework, with the objective to deal with non-random missing outcomes in an experimental setting. Their approach places very conservative bounds around treatment effect estimates. Essentially, this approach implies determining the worst- and best-case scenario for the missing outcomes, based on the observed data. By

replacing the missing outcomes with the worst- and best-case outcomes, a lower and upper bound is constructed for the estimated treatment effect $\beta$. While this approach provides an intuitive benchmark, a major disadvantage is that these conservative bounds in clinical trials suffering from sizable attrition can often turn out to be largely uninformative, as they tend to be very wide and (thus) often include zero. Yet, this bounds analysis approach is valuable in being among the first to (1) acknowledge the potential severity of the imposed bias on the point estimate due to missing outcome data, and (2) provides an intuitive solution for dealing with this uncertainty by focusing on bound estimates for $\beta$, rather than on a single point estimate.

By imposing a monotonicity assumption on the selection mechanism of missing outcomes, Lee [12] provided a -generally- more useful bounds procedure which generates relatively tighter bounds. This assumption entails that assignment to treatment can only affect the likelihood of attrition in one direction, such that there is no heterogeneous effect–in terms of sign- of treatment assignment on attrition.

Let $S_i$ be a binary indicator, with $S_i = 0$ indicating that outcome $y_i$ is not observed due to attrition. Furthermore, let $1(\cdot)$ be the indicator function which counts the number of observations satisfying status indicated between parenthesis. The proportion of outcomes observed for treatment (T = 1) and control (T = 0) group can then be written as:

$$q_{T=1} = \frac{\sum_i 1 \cdot (T_i = 1, S_i = 1)}{\sum_i 1 \cdot (T_i = 1)}$$
$$q_{T=0} = \frac{\sum_i 1 \cdot (T_i = 0, S_i = 1)}{\sum_i 1 \cdot (T_i = 0)}.$$

(7)

The bounds procedure first determines which group suffers less from attrition. In accordance with the empirical example presented later in this section, and without loss of generalization, we assume that $q_{T=0} > q_{T=1}$. The Lee bounds procedure then trims the outcome distribution of the control group by removing $q = \frac{q_{T=0} - q_{T=1}}{q_{T=0}}$ observations from the lower (upper) end of the distribution such that an upper (lower) bound for the treatment effect $\beta$ is estimated. Intuitively, the $q$-value indicates the proportion of control observations that ought to be removed in order to achieve a similar attrition rate in both the control and treatment group.

If the monotonicity assumption of sample selection holds, the treatment effect bounds estimated apply to so-called *never-attriters*. These are participants for whom the outcome will be observed ($S_i = 1$), irrespective of being assigned to the treatment (T = 1) or control (T = 0) group. By virtue of randomization, the treatment control difference in means will then provide bias-corrected bounds estimates of the average treatment effect for this particular group of participants. This subpopulation can be characterized by the distribution of **X** for participants with non-missing outcome data in the assignment group that remains untrimmed (here: treatment). Also, when the trimming proportion is zero, there is a limited test of whether the monotonicity assumption holds for observables (Lee, 2009) by estimating Eq 4 on the subsample of participants selected in the bounding procedure and by verifying that treatment status $T_i$ cannot be predicted by **X**. The monotonicity assumption is the only identifying assumption, and if violated the estimated lower and upper bound may provide biased bounds estimates for the never attriters.

**Empirical example.**   To showcase the Lee bounds procedure, the randomized clinical trial data from Donker et al. (2019) is utilized. The objective of this study was to estimate the treatment effect of ZeroPhobia—a virtual reality cognitive behavior therapy (VR CBT)–on acrophobia symptoms. In a single-blind randomized clinical trial, 193 participants were randomly

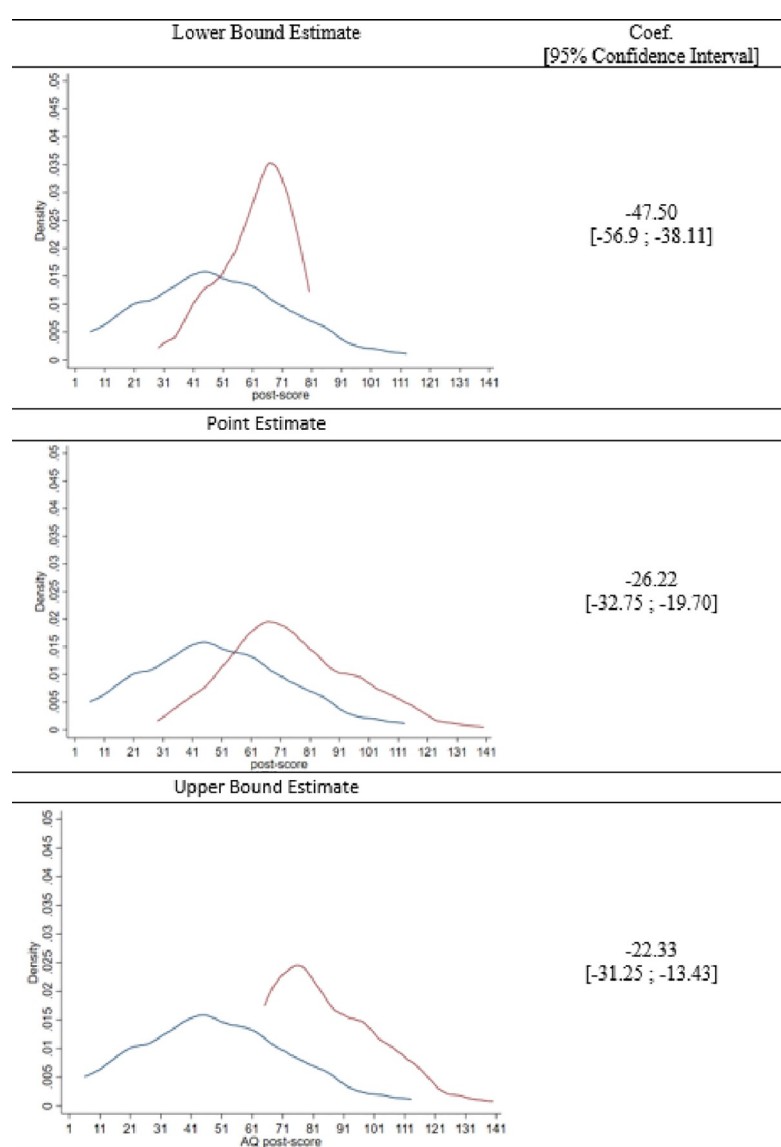

**Fig 1. Lee bounds trimming procedure.**

assigned to the treatment (n = 96) and a wait-list control group (n = 97). To measure acrophobia the Acrophobia Questionnaire was used [13] and the mean (SD) AQ-Total outcome scores for the treatment and control group were 48.46 (24.33) and 74.68 (21.55), respectively.

For 11 (40) participants of the control (treatment) group, no outcome was observed. The attrition proportion is therefore largest for the wait-list control group, such that the $q = \frac{q_{T=0} - q_{T=1}}{q_{T=0}} = \frac{(86/97) - (56/96)}{(86/97)} = 0.34$ (or 34 percent). To generate a lower and upper bound estimate for treatment effect $\beta$, the Lee bounds procedure then trims the outcome distribution of the wait-list control group by removing consecutively the 34 percent highest and lowest outcome values observed.

Fig 1 visualizes the Lee bounds, when applied to the data of Donker et al. [14]. Fig 1 shows three panels in which the blue line represents the untrimmed intervention distribution and the red line the distribution of the control group, which is either trimmed (i.e. the upper and

lower panel) or untrimmed (i.e. the middle panel). The middle panel shows the treatment effect point estimate as a reduction in acrophobia symptoms with 26.22 points (i.e. 74.68–48.46), when potential attrition bias as a result of MNAR is not taken into account. The distributions associated with the lower bounds estimate display that the upper 34 percent of the control distribution is trimmed (removed), yielding a lower bound estimate of -22.33. Moreover, the confidence interval of this lower bound estimate indicates that the mean difference of the AQ post-score is significantly smaller than zero. This is an important empirical result, as it indicates that even the most conservative Lee bound estimate indicates that ZeroPhobia significantly reduced acrophobia symptoms. The upper bound estimate of -47.50 points is obtained by trimming the lower 34 percent of the control distribution. The Lee bound results thus indicate that the estimated treatment effect is arguably somewhere between this interval of -22.33 and -47.50 points. When the confidence intervals for the point estimates of both bounds are also taken into consideration, the corresponding interval estimate for the average treatment effect is [-13.43; -56.90].

The Lee bound procedure can also be performed conditionally on discrete background characteristic variables, with the objective to further tighten the bounds thus obtained. This conditioned approach essentially implies that the trimming is performed separately for each category as defined by the discrete variable(s). In order to illustrate this 'tightening by conditioning'- effect, suppose that only gender is used as the conditioning variable and that attrition occurs solely among female participants. In the unconditional bounding procedure, all extreme values of the control group outcome distribution are trimmed, regardless gender. In the conditional version, however, extreme observations of male participants in the control group are not trimmed (i.e. only extreme observations of women are). If we would trim only the 34 percent for women in the control group, then the conditional and unconditional Lee bounds are similar only if these trimmed female observations are all located at the extremes of the distribution. If this is not the case, the bounds will be relatively tight, as the resulting lower and the upper bound estimates will be closer around the initial point estimate of $\beta$.

### Random forest lee bounds

When assuming MNAR and estimating corresponding bounds for the average treatment effect, a limitation of the aforementioned conditioning procedure with Lee bounds is that it requires discrete groups to perform the trimming procedure. As such, the set of background characteristics that can be used to tighten the bounds is restricted to include only non-continuous variables. Thus, if attrition can to a large extend be explained by continuous variables, this information can unfortunately not be optimally exploited by the Lee bounds procedure. Therefore, we propose a Random Forest Lee Bounds (RFLB) approach, such that also continuous variables can be used for obtaining tighter bounds around the estimate for treatment effect $\beta$.

The RFLB procedure first classifies data points efficiently into the attrition class they belong to. For this, a decision tree is used together with an entropy ($E$) function. The entropy function measures the purity of the data and -in the context of attrition- this indicates the proportion of attrition. The decision tree is used to partition data recursively into two groups such as to maximize data purity. This recursive data-splitting process can be explained using Fig 2, under the simplifying assumption that the observed input vector **X** consists of only two background characteristics, $x_1$ and $x_2$ (this example is taken from Plak et al. [15]). The red dots in the figure represent $\{x_1, x_2\}$ observations for missing outcomes, while the green dots represent non-missing outcome observations. In this example, the first algorithmic split occurred at $x_2 \geq a_2$, implying that the two data samples obtained by this split are more pure than the initial data

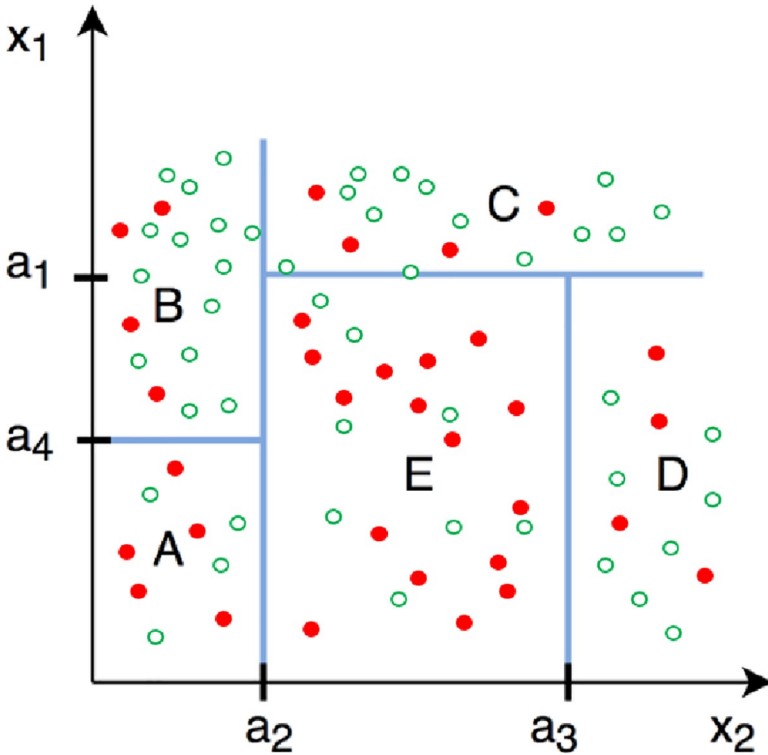

**Fig 2. Decision tree data splitting.** This is a revised figure, taken from Zou & Schonlau [17].

sample. The second, third and fourth split occurred at $x_1 \geq a_4$, $x_1 \geq a_1$, and $x_2 \geq a_3$, respectively.

The algorithm stops when splitting the data does not yield more purified data samples and avoids further reduction of degrees of freedom for the adjusted residual sum of squares. We note that the random forest approach can be applied using multiple conditioning variables and is suitable for both categorical classification problems (such as shown in Fig 2), and continuous prediction problems. The example explained above can also be represented as a continuous outcome problem for which the random forest approach builds regression trees instead of classification trees and indicates–in our case—which characteristics best explain the observed variation in attrition.

The random forest approach avoids over-fitting by applying k-fold cross-validation and takes into account that small changes in the data can yield large changes in the final tree obtained, by selecting the splitting variable at each step from *m* out of *x* randomly drawn input variables [16].

The Random Forest Lee Bounds (RFLB) procedure presented here can be summarized in four steps. First, a random forest is estimated using a set of background characteristics. Second, an importance graph is created which indicates how effective each characteristic reduces the variance (i.e. increases the prediction performance of the model). A random benchmark variable is included as covariate in the model, such that it can be evaluated if a covariate is more important than a random benchmark variable and–thus- whether it should be used to tighten the bounds. Third, a decision tree is estimated with only those variables included that were marked as important by the importance graph in the previous step. Fourth, discrete group indicator variables are generated using the decision tree from the previous step and used as conditioning variables in the Lee bounds procedure.

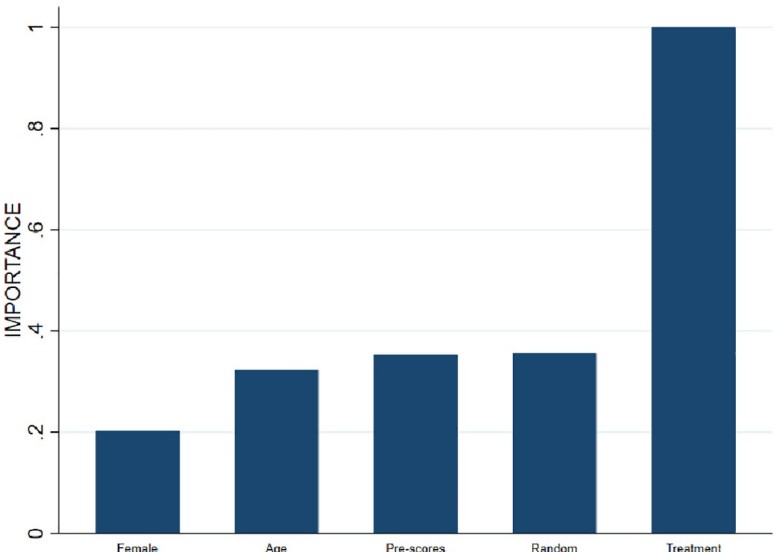

**Fig 3. Importance graph attrition variation.**

To reiterate, the rationale for using the random forest approach is to ensure that also continuous variables can be optimally exploited in the bounding procedure, and not to optimize the out-of-sample predictive power of the model. The obtained bounds will be tighter if these continuous variables statistically explain attrition. To see whether this is the case, random forest lee bounds are estimated applying the aforementioned procedure and using the ZeroPhobia randomized clinical trial data. The importance graph presented by Fig 3 indicates that the treatment indicator variable explains most of the attrition variation. This variable therefore receives an importance value of 1 and all other variables receive a value relative to that of the most important variable. The random variable receives a value of .3 and–importantly- all other variables (i.e, gender, age and AQ pre-scores) are considered less important in explaining variance than this random variable. This implies that the association between attrition and the variables gender, age and AQ pre-scores is not statistically significant and explains why Donker et al. [14] perform an unconditional Lee bounds procedure. Yet, if the covariation between attrition and treatment assignment is indeed associated with some background characteristics, then the bounds can be tightened by conditioning on these variables.

To illustrate that additional conditioning on a continuous variable can further tighten the bounds, attrition is simulated in the data from Donker et al. [14], such that the selective nature of the simulated attrition is now associated jointly with treatment status, gender and pre-score, while leaving the average treatment effect estimate unaffected. The simulation method is outlined in more detail in S2 Appendix. Fig 4 shows the importance graph and illustrates that the attrition is selective and conditional on the simulated characteristics gender and pre-scores.

That attrition is selective with respect to gender and pre-score is a necessary but not sufficient condition for the bounds to be tightened by RFLB. It is only sufficient when the covariance between attrition and treatment assignment is conditional on these background characteristics as well. Fig 5 shows that these interaction terms are indeed important, which is why the variables Female and AQ pre-scores are selected for conditioning.

Table 1 summarizes the various bounding estimates obtained when performing unconditional Lee bounds, conditional Lee bounds and the RFLB introduced here. Columns 2 and 3 show the estimated lower and upper bound. Columns 4 and 5 show the 95% confidence

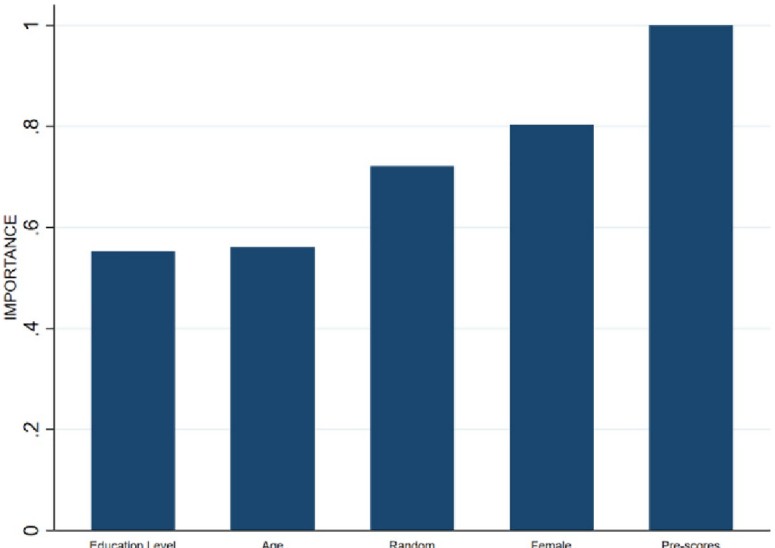

**Fig 4. Importance graph simulated attrition.** The relative importance of the intervention variable is excluded from the figure, as only background characteristics are to be considered for tightening Random Forest Lee Bounds.

intervals, which can be considered a more appropriate comparison between the different bounding approaches, as this also acknowledges precision loss resulting from the reduction of degrees of freedom when conditioning on background characteristics.

The estimation results of Table 1 show that the intervals are most tight when RFLB is performed. The 95% confidence intervals indicate that–relative to unconditional Lee bounds- the reduction in interval width is 7.2% with conditional Lee bounds, and 26.6% with RFLB. It follows that RFLB is the preferred bounding approach, as it provides the most precise bias-corrected interval estimate for the average treatment effect of ZeroPhobia.

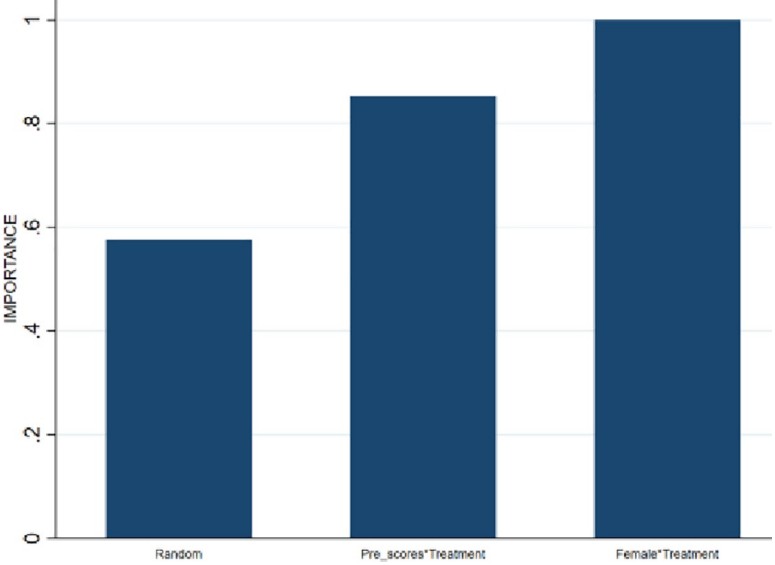

**Fig 5. Importance graph simulated attrition interaction terms.**

**Table 1. Bounding estimation results.**

| | Estimated Treatment Effect | | | |
| --- | --- | --- | --- | --- |
| | | | 95% conf. interval | |
| | Lower | Upper | Lower | Upper |
| Unconditional Lee- bound | -37.33 | -17.89 | -39.86 | -15.44 |
| | *Width*: 19.44 | | *Width*: 24.42 | |
| Conditional Lee bound | -37.70 | -19.85 | -40.15 | -17.49 |
| (Female) | *Width*: 17.85 | | *Width*: 22.66 | |
| Random forest Lee bound | -32.65 | -20.39 | -35.86 | -17.94 |
| (Female, Pre-scores) | *Width*:12.26 | | *Width*:17.92 | |

To generalize the bounding estimation results we also performed simulations in which the bounding estimates were again obtained under different treatment sizes: small, medium and large (i.e. $d = 0.2$, $d = 0.4$ and $d = 0.8$). These bounding results are shown in Table 2 and the results suggest that the unconditional Lee bounds are tightened by the RFLB approach with 19.8, 16.8 and 16.6 percent for respectively a small, medium and large treatment size. This is an important result: when the effect size becomes smaller it is more likely that 0 lies and thus more important to obtain tighter bounds.

More generally, it holds that RFLB will always produce interval widths that are smaller or similar compared to the other bounding approaches *if* the importance graph indicates that the interaction terms explain a significantly amount of the in attrition. This result is intuitive, since both conditional bounding approaches ensure that similar or less extreme values of the observed outcome distributions are trimmed, resulting tighter interval estimates. Moreover, the imposed assumptions by RLFB are equivalent to the Lee bounds procedure, but RFLB allows for conditioning on continuous variables, which is crucial for precision gains in randomized clinical trials as the pre-treatment scores is–often- the best predictor of the outcome observed at end point.

## Discussion

This study proposes to use a new Random Forest Lee Bounds (RFLB) approach that can be applied to estimate relatively tight average treatment effect intervals, while allowing for non-

**Table 2. Bounding estimation results for small, medium and large effect sizes.**

| | Lower | Upper | Diff. | %-gain | 95% conf. interval Lower | Upper | Diff. | %-gain |
| --- | --- | --- | --- | --- | --- | --- | --- | --- |
| **Unconditional Lee Bound** | | | | | | | | |
| Small (d = 0.2) | -0.280 | -0.162 | 0.118 | . | -0.297 | -0.145 | 0.153 | . |
| Medium (d = 0.4) | -0.560 | -0.325 | 0.235 | . | -0.595 | -0.289 | 0.306 | . |
| Large (d = 0.8) | -1.119 | -0.650 | 0.470 | . | -1.189 | -0.578 | 0.610 | . |
| **Conditional Lee Bound** (Female) | | | | | | | | |
| Small (d = 0.2) | -0.281 | -0.164 | 0.117 | 0.825 | -0.298 | -0.147 | 0.152 | 0.655 |
| Medium (d = 0.4) | -0.557 | -0.326 | 0.231 | 1.836 | -0.591 | -0.291 | 0.300 | 1.768 |
| Large (d = 0.8) | -1.123 | -0.644 | 0.479 | -1.961 | -1.193 | -0.573 | 0.620 | -1.540 |
| **Random Forest Lee Bound** (Female, Pre-score) | | | | | | | | |
| Small (d = 0.2) | –0.269 | -0.175 | 0.094 | 19.818 | -0.287 | -0.155 | 0.132 | 13.294 |
| Medium (d = 0.4) | -0.533 | -0.338 | 0.196 | 16.802 | -0.570 | -0.297 | 0.272 | 10.933 |
| Large (d = 0.8) | -1.069 | -0.677 | 0.392 | 16.609 | -1.142 | -0.597 | 0.545 | 10.665 |

random missing outcome data (MNAR). An appealing feature of the RFLB procedure is that it allows for tightening intervals by exploiting the observed associations of attrition with both categorical *and* continuous background characteristics, as has been demonstrated empirically in this study.

In the case of a randomized clinical trial, the internal validity of RFLB entails an additional identifying monotonicity assumption. In the case of randomized clinical trials, this implies that if–for example- assignment to treatment induces some participants to drop out, it cannot simultaneously cause other participants to *no*t drop out. If this assumption holds, RFLB results are internally valid for the subpopulation of *non-attriters*.

Missing outcome data are unavoidable in randomized clinical trials and arguably depend on non-random unobserved data [18]. Bounding approaches serve to accommodate such selective attrition processes and to acknowledge the resulting uncertainty with respect to causal inference. Yet, the potential of bounding procedures for generating informative–relatively precise- interval estimates for the average treatment effect depends crucially on the ability to statistically predict attrition using pre-treatment characteristics.

## Conclusion

The importance of randomization in clinical trials has long been acknowledged for avoiding selection bias and enabling internal validity of the research results reported. Yet, many clinical trials suffer from attrition between randomization and follow-up data collection, such that attrition bias reintroduces concerns with respect to causal inference. This study takes a causal inference perspective when addressing distinct scenarios of missing outcome data (MCAR, MAR and MNAR) and provides an overview of empirical strategies that can be applied to estimate the average treatment effect, improve precision of the estimator, and to test whether the underlying identifying assumptions hold.

While many studies assume MCAR or MAR, the corresponding identifying assumptions are very restrictive, often untenable [18], and can hardly be verified empirically using the observed data. It is therefore appropriate to allow missing outcome data in clinical trials to be the result of non-random unobserved events (i.e. MNAR) and to address this issue accordingly. Yet, existing empirical approaches for dealing with MNAR rely on exploring worst-case scenario bounds and often result in very wide, uninformative, treatment effect estimate intervals.

By exploiting information from all observed background characteristics, the RFLB procedure proposed in this paper can considerably increase the statistical power of bias-corrected treatment effect estimates for randomized clinical trials facing non-random missing outcome data. As such, RFLB can hopefully facilitate researchers -whether involved in clinical trials or other disciplines- in allowing for post-assignment attrition to be non-random in nature and in addressing this accordingly.

## Supporting information

**S1 Appendix.**
(DOCX)

**S2 Appendix.**
(DOCX)

**S1 Data.**
(DTA)

**S2 Data.**
(DO)

## Author Contributions

**Methodology:** Ilja Cornelisz, Chris van Klaveren.

**Writing – original draft:** Ilja Cornelisz, Chris van Klaveren.

**Writing – review & editing:** Pim Cuijpers, Tara Donker.

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
