## [Decision Letter · Decision Letter 0]

1 May 2020

PONE-D-20-02997

Addressing Missing Data in Randomized Clinical Trials: A Causal Inference Perspective

PLOS ONE

Dear PhD van Klaveren,

Thank you for submitting your manuscript to PLOS ONE. After careful consideration, we feel that it has merit but does not fully meet PLOS ONE’s publication criteria as it currently stands. Therefore, we invite you to submit a revised version of the manuscript that addresses the points raised during the review process.

I agree with the reviewers, and would say minor revision.  But I do have one other comment to add.  Specifically, this statement here:

In other words, randomization enables internal validity (Rubin, 1974), such that

differences between participants in a clinical trial assigned to different treatment arms can,

apart from random error, be attributed to the treatment under investigation (Jüni, Altman &

Egger, 2001).

This claim is categorically untrue.  The reason for it being false should be discussed, as opposed to just deleting it, since this is a common sentiment, and needs to be refuted.  It is somewhat disgraceful that authors are still making this claim even today, when we know better.  I would refer the authors to my book:

Berger, VW (2005). “Selection Bias and Covariate Imbalances in Randomized Clinical Trials”, John Wiley & Sons, Chichester.

We would appreciate receiving your revised manuscript by Jun 15 2020 11:59PM. To enhance the reproducibility of your results, we recommend that if applicable you deposit your laboratory protocols in protocols.io, where a protocol can be assigned its own identifier (DOI) such that it can be cited independently in the future. For instructions see: http://journals.plos.org/plosone/s/submission-guidelines#loc-laboratory-protocols

We look forward to receiving your revised manuscript.

Kind regards,

Vance Berger

Academic Editor

PLOS ONE

Journal Requirements:

Additional Editor Comments (if provided):

I do have one other comment to add. Specifically, this statement here:

In other words, randomization enables internal validity (Rubin, 1974), such that

differences between participants in a clinical trial assigned to different treatment arms can,

apart from random error, be attributed to the treatment under investigation (Jüni, Altman &

Egger, 2001).

This claim is categorically untrue. The reason for it being false should be discussed, as opposed to just deleting it, since this is a common sentiment, and needs to be refuted. It is somewhat disgraceful that authors are still making this claim even today, when we know better. I would refer the authors to my book:

Berger, VW (2005). “Selection Bias and Covariate Imbalances in Randomized Clinical Trials”, John Wiley & Sons, Chichester.

Reviewers' comments:

Reviewer's Responses to Questions

**Comments to the Author**

1. Is the manuscript technically sound, and do the data support the conclusions?

Reviewer #1: Yes

Reviewer #2: Yes

2. Has the statistical analysis been performed appropriately and rigorously? 

Reviewer #1: Yes

Reviewer #2: Yes

3. Have the authors made all data underlying the findings in their manuscript fully available?

Reviewer #1: No

Reviewer #2: Yes

4. Is the manuscript presented in an intelligible fashion and written in standard English?

Reviewer #1: Yes

Reviewer #2: Yes

5. Review Comments to the Author

Reviewer #1: The paper addresses an important topic - missing data and related causal inference in RCTS.

The paper is well-written but could be improved further with more relevance to readers with the inclusion of (1) simulations with different baseline covariate distributions under different treatment effect sizes: small, medium and large and (2) relevant data and syntax to enable replication of authors findings.

Reviewer #2: The manuscript by Cornelisz et al. takes a causes inference approach to address selection bias and missingness in outcomes due to MCAR, MAR, and MNAR in clinical trials. To do this, the authors provide causal inference-based strategies for estimating the average treatment effect, improve precision, and to test some underlying assumptions related to missingness.

The article was well written and easy to follow.

See below for my comments.

\\begin{enumerate}

\\item On page 10, change "demonstrate" to "demonstrated" in the sentence "The potential of exploiting RFLB...".

\\item On page 11, the sentence "Individuals are randomly assigned to one of both treatment conditions, such that always only one of both potential outcomes...." seems a little confusing. Can the authors clarify this sentence?

\\item On page 15, the sentence "The term 'missing at random' therefore does not defines missing outcomes as random events" also needs to be clarified. Can the authors double check this sentence?

\\item On page 17, can the authors say more about the monotonicity assumption since the assumption of the bounding procedure depends on this. What happens when the monotonicity assumption is violated?

\\item Are there certain assumptions/conditions that need to be met in order to apply the RFLB - this was not clear. Can you say more about the assumptions?

\\end{enumerate}

\\end{document}

6. PLOS authors have the option to publish the peer review history of their article (what does this mean?). If published, this will include your full peer review and any attached files.

Reviewer #1: No

Reviewer #2: No

---

## [Author Response · Author response to Decision Letter 0]

21 May 2020

We have responded to the reviewers and the editor in the uploaded documents (cover letter and reviewer comments). We note that we did not used track changes, but presented this text and the placement also in the uploaded reply.

---

## [Editor Report · Decision Letter 1]

26 May 2020

Addressing Missing Data in Randomized Clinical Trials: A Causal Inference Perspective

PONE-D-20-02997R1

Dear Dr. van Klaveren,

We are pleased to inform you that your manuscript has been judged scientifically suitable for publication and will be formally accepted for publication once it complies with all outstanding technical requirements.

With kind regards,

Vance Berger

Academic Editor

PLOS ONE
---

## [Editor Report · Acceptance letter]

5 Jun 2020

PONE-D-20-02997R1 

Addressing Missing Data in Randomized Clinical Trials: A Causal Inference Perspective 

Dear Dr. van Klaveren:

I'm pleased to inform you that your manuscript has been deemed suitable for publication in PLOS ONE. Congratulations! Your manuscript is now with our production department. 

Kind regards, 

on behalf of

Dr Vance Berger 

Academic Editor

PLOS ONE